# A Novel 65-bp Indel in the *GOLGB1* Gene Is Associated with Chicken Growth and Carcass Traits

**DOI:** 10.3390/ani10030475

**Published:** 2020-03-12

**Authors:** Rong Fu, Tuanhui Ren, Wangyu Li, Jiaying Liang, Guodong Mo, Wen Luo, Danlin He, Shaodong Liang, Xiquan Zhang

**Affiliations:** 1Department of Animal Genetics, Breeding and Reproduction, College of Animal Science, South China Agricultural University, Guangzhou 510642, China; furong325@stu.scau.edu.cn (R.F.); TuanhuiRen@foxmail.com (T.R.); liwangyu224@stu.scau.edu.cn (W.L.); jyliang@163.com (J.L.); mgd9527@163.com (G.M.); luowen729@scau.edu.cn (W.L.); dlhe@scau.edu.cn (D.H.); sdliang@scau.edu.cn (S.L.); 2Guangdong Provincial Key Lab of Agro-Animal Genomics and Molecular Breeding, and Key Laboratory of Chicken Genetics, Breeding and Reproduction, Ministry of Agriculture, Guangzhou 510642, China

**Keywords:** chicken, *GOLGB1*, indel, growth and carcass traits

## Abstract

**Simple Summary:**

Many Chinese-local chickens show slow-growing and low-producing performance, which is not conductive to the development of the poultry industry. The identification of thousands of indels in the last twenty years has helped us to make progress in animal genetics and breeding. Golgin subfamily B member 1 (*GOLGB1*) is located on chromosome 1 in chickens. Previous study showed that a large number of QTLs on the chicken chromosome 1 were related to the important economic traits. However, the biological function of *GOLGB1* gene in chickens is still unclear. In this study, we detected a novel 65-bp indel in the fifth intron of the chicken *GOLGB1* gene. Correlation analysis between the 65-bp indel and chicken growth and carcass traits was performed through a yellow chicken population, which is commercial. Results revealed that this 65-bp indel was significantly associated with chicken body weight, highly significantly associated with neck weight, abdominal fat weight, abdominal fat percentage, and the yellow index b of breast. These findings hinted that the 65-bp indel in *GOLGB1* could be assigned to a molecular marker in chicken breeding and enhance production in the chicken industry.

**Abstract:**

Golgin subfamily B member 1 (*GOLGB1*) gene encodes the coat protein 1 vesicle inhibiting factor, giantin. Previous study showed that mutations of the *GOLGB1* gene are associated with dozens of human developmental disorders and diseases. However, the biological function of *GOLGB1* gene in chicken is still unclear. In this study, we detected a novel 65-bp insertion/deletion (indel) polymorphism in the chicken *GOLGB1* intron 5. Association of this indel with chicken growth and carcass traits was analyzed in a yellow chicken population. Results showed that this 65-bp indel was significantly associated with chicken body weight (*p* < 0.05), highly significantly associated with neck weight, abdominal fat weight, abdominal fat percentage and the yellow index b of breast (*p* < 0.01). Analysis of genetic parameters indicated that “*I*” was the predominant allele. Except for the yellow index b of breast, *II* genotype individuals had the best growth characteristics, by comparison with the *ID* genotype and *DD* genotype individuals. Moreover, the mRNA expression of *GOLGB1* was detected in the liver tissue of chicken with different *GOLGB1* genotypes, where the *DD* genotype displayed high expression levels. These findings hinted that the 65-bp indel in *GOLGB1* could be assigned to a molecular marker in chicken breeding and enhance production in the chicken industry.

## 1. Introduction

Golgin subfamily B member 1 *(GOLGB1)* gene is not only a widely expressed large coiled-coil protein, but also a Golgi-associated large transmembrane protein [1]. To date, mutations in Golgi-associated proteins have been found to be associated with scores of human developmental disorders and diseases [2,3]. A previous study reported that the *GOLGB1* loss-of-function mutation was co-segregated with cleft palate, and *GOLGB1* mutant embryos showed intrinsic defects in palatal shelf elevation. These results suggest that *GOLGB1* plays an important role in glycosylation and tissue morphogenesis of protein [4]. Another previous study reported that *GOLGB1* gene is expressed in cultured chondrocytes, and various organs and embryos during different developmental stages [5]. It was also found that the coat protein 1 vesicle tethering factor encoded by the *GOLGB1* gene plays a key role in many aspects of cartilage [6]. A 10-bp indel in exon 13 of the *GOLGB1* gene in OCD rats is associated with spontaneous osteochondral dysplasia and systemic edema [6]. *GOLGB1* rs3732410 was found to be significantly associated with a reduced risk of hemorrhagic stroke (HS) in people ≤ 50 years of age [7]. A genome-wide associated study has shown that one mutation in *GOLGB1* (Y1212C) was significantly associated with a lower risk of stroke [8].

Many Chinese-local chickens show slow-growing and low-producing performance, which is not conductive to the development of the poultry industry, such as the Xinghua chickens, Qingyuan Partridge chickens, Lushi chickens, Gushi chickens, and Wenchang chickens in this study. The identification of thousands of indels in the last twenty years helped us to make progress in animal genetics and breeding. Mills et al. created the first indel map of the human genome, and an indel is a small genetic variation between 1 and 10,000 base pairs in length [9]. The number of indels in the genome is second only to that of single nucleotide polymorphisms (SNPs). Through whole-genome sequencing of chickens, indels were found to play an important role in genetic diversity and phenotypic divergence [10,11]. During the whole-genome resequencing of domestic chicken leg feather traits, more than 2.1 million short indels were obtained [12]. Previous study has detected and identified about 883,000 high quality indels by whole-genome analysis of several modern layer chicken lines from diverse breeds [13]. Therefore, understanding indels in great detail is therefore important for profiling genetic variation within the genome, studying the evolution relationship of species, and detecting casual mutations of genetic disorders [13]. A large number of researches have elucidated the effects of indels’ polymorphism on livestock production in other gene structures [14]. For example, a 16-bp indel in *KDM6A* was significantly associated with growth traits in goat [15], and a 10-bp indel in *PAX7* was associated with growth traits in cattle [16]. Many indels of functional genes in chicken were reported to associate with different chicken phenotypes. A 9-bp indel polymorphism in *PMEL17* was associated with plumage color [17], a 24-bp indel in *PRL* was associated with egg production [18], a 8-bp indel in *GHRL* was associated with growth [19], a 31-bp indel in *PAX7* was associated with performance [20], a multiallelic indel in the promoter region of *CDKN3* gene was associated with body weight and carcass traits in chickens [21], a 80-bp indel in *PRLR* was associated with chicken growth and carcass traits [22], a 86-bp indel in *MLNR* was associated with chicken growth [23], two indels in the *QPCTL* gene were associated with body weight and carcass traits in chickens [24], and a 62-bp indel in *TGFB2* was associated with body weight [25].

In our previous study identifying candidate genes underlying chickens’ yellow skin with resequencing data from Genbank, the *GOLGB1* gene is found to be located on chromosome 1 and might be potentially associated with the yellow skin phenotype (data not published). As the candidate gene underlying yellow skin may also be associated with chicken growth, we screened the variations of the gene using the resequencing data from Chinese-local chicken breeds and Recessive White Rock chickens with various growing rates. As indicated above, this gene is well-studied in humans, but not in chickens. Furthermore, a total of 188 indels were identified in the *GOLGB1* gene of chickens (https://ensembl.org/Gallus_gallus/Gene/Variation_Gene/Table?align=1760; db=core;g=ENSGALG00000041494; r=1:323123–357594). However, there is no report about the indel function of chickens on the *GOLGB1* gene. Previous study has shown that a large number of quantitative trait locis (QTLs) on chicken chromosome 1 are related to the important economic traits such as growth [26]. However, the function of *GOLGB1* on chicken growth is unclear. In this study, we detected a 65-bp indel in the fifth intron of *GOLGB1* gene among Chinese-local chicken breeds. We further studied the relationship between the 65-bp indel in *GOLGB1* and chicken growth traits, and further observed its expression patterns in different tissues. These results may provide a theoretical basis for further research on the application of molecular marker technology in the chicken industry.

## 2. Materials and Methods

All animal experiments in this study were conducted in accordance with the protocols approved by the South China Agriculture University Institutional Animal Care and Use Committee (approval number: SCAU#0015) and also in accordance with the Animal Protection Law of the People’s Republic of China.

### 2.1. Animal Samples and Genomic DNA Collection

In total, 1358 chicken samples, were composed of eight different Chinese-local chicken breeds, including Tianlu yellow chicken (N409, *n* = 382, 13 w of age), Mahuang chickens (MH, *n* = 578, 12 w of age), Wenchang chickens (WC, *n* = 88, 7 w of age), Guangxi Sanhuang chickens (SH, *n* = 72, 12 w of age), Gushi chickens (GS, *n* = 69, 16 w of age), Xinghua chickens (XH, *n* = 71, 17 w of age), Qingyuan Partridge chickens (QY, *n* = 60, 7 w of age), and Lushi chickens (LS, *n* = 38, 6 w of age). The N409 population was from Guangdong Wen’s Southern Poultry Breeding Co., Ltd. in Guangdong Province, China. A total of 18 traits were recorded, including carcass traits, body size traits, and fatness traits, and were used for statistical analysis [27].

DNA samples in this study were extracted from blood samples by using the NRBC Blood DNA Kit (OMEGA, BIO-TEK, Vernuski, VT, USA), following the manufacturer’s instructions. After using a Nanodrop2000c spectrophotometer (Thermo Scientific, Waltham, MA, USA) to assay, all DNA samples were uniformly diluted to 80 ng/μL and stored at −20 °C.

In addition, 10 Xinghua chickens and 10 Recessive White Rock chickens were used for whole-genome sequencing using Hiseq 2500. The average sequencing coverage of the two lines was 10X. High-quality sequencing libraries were constructed by stringently following the standard protocol of IlluminaTruSeq™ DNA preparation kit (Illumina, San Diego, CA, USA).

### 2.2. Genetic Variation and Genotyping

In order to get more information about the polymorphism distribution of the 65-bp indel, we used PCR amplification and agarose gel electrophoresis to detect the genotypes in the eight different Chinese-local chicken breeds. According to the *GOLGB1* gene sequence published on NCBI, the primer P1 (Table 1) was designed to amplify the DNA fragment including the 65-bp indel. A 10.0 μL reaction system was set up for each sample, containing 1.0 μL of template DNA, 0.3 μM per primer, 5.0 μL of 2 × *Taq* PCR StarMix (GenStar, Beijing, China), and 2.9 μL of ultrapure water. The PCR amplification procedure is as follows, pre-denaturation at 94 °C for 3 min, denaturation at 94 °C for 30 s, annealing at 60 °C for 30 s, extension at 72 °C for 30 s, 34 cycles, and finally extension at 72 °C for 5 min. Next, PCR products were detected by 2.0% agarose gel electrophoresis, and the products of different genotypes were verified by sequencing (Sangon Biotech, Shanghai, China).

The genotype and allele frequencies of the eight breeds were calculated and the Hardy–Weinberg equilibrium (HWE) was calculated by using the SHEsis program [28]. In addition, the populations indexes such as effective allele numbers (Ne), polymorphic information content (PIC), homozygosity (Ho), and heterozygosity (He) were calculated following Nei’s methods [29], which were performed by PopGene (version 1.3.1, Edmonton, AB, Canada).

### 2.3. RNA Isolation and cDNA Synthesis

RNA was isolated from thirteen tissues including heart, liver, spleen, lung, kidney, duodenum, intestine, ovary, abdominal, breast muscle, leg muscle, hypothalamus, and cerebellum of 4 MH chickens (20 w of age). The RNA was then used as a tissue expression pattern analysis after reverse transcription into cDNA. Liver tissues from 12 indigenous XH chickens were used to compare the relative mRNA expression levels of different genotypes in the *GOLGB1* gene. According to the manufacturer’s instructions, the total RNA in each tissue was extracted using Trizol reagent (TaKaRa, Otsu, Japan). The integrity of all obtained RNA samples were determined by 1.2% agarose gel electrophoresis, and the concentrations of all RNA samples were measured by a Nanodrop2000c spectrophotometer (Thermo Scientific, Waltham, MA, USA).

The PrimeScript RT Reagent Kit (Perfect Real Time) (TaKaRa, Otsu, Japan) was used for the complementary DNA (cDNA) synthesis of mRNA according to the instructions of the manufacturer.

### 2.4. Real-Time Quantitative PCR Analysis

According to the manufacturer’s instructions, real-time quantitative PCR (qRT-PCR) was performed on an ABI Quantstudio 5 Real-Time PCR System (ABI, Los Angeles, CA, USA) using an iTaqTM Universal SYBR^®^ Green Supermix Kit (Bio-Rad, Hercules, CA, USA). Three repetitions for each sample were designed to make the result more accurate. Primer Premier5.0 software (Premier Bio-soft International, Palo Alto, CA, USA) was used to design the qPCR primers, then sent to Tsingke Biotech Co. Ltd. (Guangzhou, China) for synthesis, and *β-actin* was used to internally reference genes. Primer information is shown in Table 1. The 2^−ΔΔ*C*T^ method was used to calculate the relative mRNA expression of the *GOLGB1* gene. After the data was normalized, significance was determined by ANOVA. Data were presented as the mean ± standard error.

### 2.5. Statistical Analysis

The association analysis was performed on the genotypes and the traits of the N409 chickens using the General Linear Models Procedures of SAS 9.0 (SAS Institute Inc., Cary, NC, USA). A mix procedure was used to analyze its genetic effects. The specific GLM model is as follows,
Yijkl = μ + Si + Gj + Hk + Fl + eijkl
where Y represents the traits’ phenotypic values; μ represents the all population mean; S represents the fixed effect of sex; G represents the fixed effect of genotype or haplotype pair; H represents the fixed effect of hatch; F represents the fixed effect of family; e represents the random residuals. The *p*-value was calculated by the Student’s *t*-test [30]. All data are expressed as mean ± standard error, too.

## 3. Results

### 3.1. Identification of a Novel GOLGB1 65-bp Indel Polymorphism

A novel 65-bp indel (NC_006088.5:g.332982instgcccagcaaaagtgaagagcctcactgagctgcccagtactcact gctgctcatcctgctggtg) was found in the fifth intron in the *GOLGB1* gene (Gene ID: 426868, Figure 1). And this indel has been submitted in the EVA database (PROJECT:PRJEB37183). Three genotypes of *II*, *ID*, and *DD* were detected in eight different Chinese-local chicken breeds, in which allele “*I*” was 311 bp in length, while allele “*D*” was 246 bp in length (Figure 2).

### 3.2. Genetic Diversity of the 65-bp Indel in the Eight Different Chinese Local Chickens

The genotypic frequencies, allele frequencies, and the diversity of the 65-bp indel in the eight different breeds were calculated and shown in Table 2. In all breeds, the frequencies of the allele “*I*” (0.51–0.77) were higher than those of the allele “*D*” (0.23–0.49). Of note, LS, as a game breed, showed the lowest frequency of allele “*I*”, and MH and N409, as improved local breeds, showed the highest frequencies of allele “*I*”. According to the result of the χ2 test, the distribution of genotypic and allele frequencies of the 65-bp indel was in Hardy–Weinberg equilibrium in all populations (*p* > 0.05). The values of He and Ne were 0.341–0.500 and 1.517–1.999, respectively. According to the classification of PIC, all of the eight Chinese-local chicken breeds exhibited moderate polymorphism at the 65-bp indel polymorphism (0.250 < PIC < 0.500), which suggested that the eight Chinese-local chicken breeds may have experienced similarly persistent selection pressures in evolutionary history.

### 3.3. Association of the 65-bp Indel with Chicken Growth Traits

As can be seen from Table 3, the 65-bp indel of the *GOLGB1* gene was significantly associated with body weight (*p* = 0.0121), neck weight (*p* = 0.0026), abdominal fat weight (*p* = 0.0006), abdominal fat percentage (*p* = 0.0015), and the yellow index b of breast (*p* = 0.0039) at 95 d of age in N409 chicken (*n* = 382). Notably, except for the yellow index b of breast, the *II* genotype is the dominant ones among the three genotypes.

### 3.4. The Expression Profiles of the Chicken GOLGB1 Gene

The mRNA expression levels of *GOLGB1* gene in 13 tissues of MH chicken were detected by qRT-PCR. The *GOLGB1* gene is expressed in all tissues. Additionally, *GOLGB1* was highly expressed in the cerebellum, hypothalamus, kidney, and abdominal fat (Figure 3).

### 3.5. Significantly Differential Gene Expression of the Three Genotypes

Statistical analysis showed that the 65-bp indel of the *GOLGB1* gene was significantly associated with growth and carcass traits, and analysis of the expression profiles showed that *GOLGB1* gene was widely expressed in the tissues of chickens. Next, we examined the expression of liver tissues in different genotypes of the *GOLGB1* gene, and found that the relative mRNA expression level of *DD* genotype was the highest. Moreover, the mRNA expression level of the *DD* genotype was significantly higher than in the *II* and *ID* genotypes (*p* < 0.01, Figure 4). These results suggested that the different genotypes might have different effects on chickens.

## 4. Discussion

Body weight, as one of the most important economic traits in the broiler-breeding industry, has been closely selected for many years [31]. As indicated above, a large number of indels associated with body weight have been reported in chickens. In this study, we first discovered a novel 65-bp indel polymorphism in the fifth intron of the *GOLGB1* gene and verified it in eight different Chinese-local breeds, and we further studied the genetic diversity and characterized the genetic properties. Finally, it was found that the 65-bp indel was significantly associated with body weight, abdominal fat weight, and abdominal fat percentage. Additionally, we found that the frequencies of the allele “*I*” was higher than those of the allele “*D*” in all breeds. And chickens with *II* genotype had significantly better body weight than those with *ID* and *DD* genotypes. Besides, we also found an interesting phenomenon that the yellow index b of breast is higher in the *DD* genotype, which is opposite to the body weight and abdominal fat weight. The relationship between abdominal fat weight and skin color needs further study.

*GOLGB1* belongs to the golgin family of large coiled-coil proteins located at the cytoplasmic surface of the Golgi apparatus [32]. *GOLGB1* is unique, because among all Golgi proteins, it contains not only a transmembrane domain that can anchor the protein on Golgi membrane or COP1 vesicle at the C-terminal, but also a p115 binding domain at the N-terminal [33,34,35,36]. The function of the *GOLGB1* gene in chicken has not been studied. In this study, we found that the 65-bp indel of intron 5 of the *GOLGB1* gene was significant associated with body weight, neck weight, abdominal fat weight, abdominal fat percentage, and the yellow index b of breast in N409 chicken. In addition, “*I*” was the predominant allele in all populations. Except for the yellow index b of breast trait, the *II* genotype was the dominant genotype in the other traits. These results indicate that the 65-bp indel may be a potential molecular marker of chicken growth and carcass traits.

Introns, non-coding spacer sequences that interrupt the linear expression of genes, are removed during the processing of the original transcription product and are not included in the sequence of mature mRNA. Introns play an important role in the regulation of gene expression [37], and a previous study has shown that removal of introns from a transgene or insertion of a transgene results in increased expression [38]. Some introns have an enhancer function [39], or contain enhancer elements that co-regulated genes with promoters [40]. In addition, SNPs on introns often alter mRNA levels by affecting transcription, RNA elongation, splicing, or maturation [41]. Cui and his colleagues identified a 16-bp indel in *KDM6A* intron 17, and found that the indel significantly affected *KDM6A* gene expression [42]. The 31-bp indel in *PAX7* intron 3 was associated with growth, carcass, and meat-quality traits of chickens [20]. However, further research is needed on its functional impact on gene expression. In this study, we found that the *GOLGB1* gene was expressed in different tissues, which is consistent with previous reports that the *GOLGB1* gene was widely expressed [1]. Additionally, the *GOLGB**1* gene was highly expressed in cerebellum, hypothalamus, kidney, liver, and abdominal fat, suggesting that it might be related to growth and fat deposition. Notably, the *GOLGB1* gene with the *DD* genotype showed higher expression than the *II* and *ID* genotypes in the liver tissue of chickens. The body weight, neck weight, abdominal fat weight, and abdominal fat percentage of individuals with the *DD* genotype are lower than those with the genotype *II*. Therefore, we argued that the 65-bp indel had a positive effect on chicken’s body weight, neck weight, abdominal fat weight, and abdominal fat percentage.

## 5. Conclusions

In conclusion, we identified a 65-bp indel in the fifth intron region of the *GOLGB1* gene. The 65-bp indel of the *GOLGB1* gene was associated with chicken body weight, neck weight, abdominal fat weight, abdominal fat percentage, and the yellow index b of breast. The above results will provide useful information for *GOLGB1* as a molecular marker for chicken-breeding programs, and enrich the understanding of the *GOLGB1* gene function.

## Figures and Tables

**Figure 1 animals-10-00475-f001:**
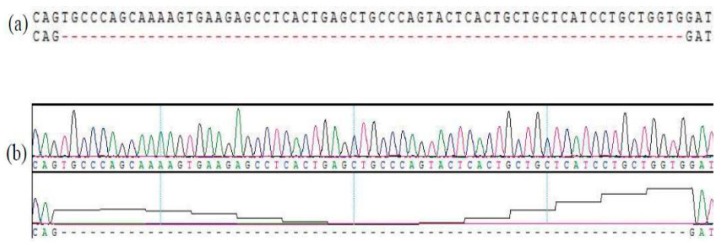
A 65-bp indel mutation in the fifth intron region of the GOLGB1 gene was identified. (**a**) Sequence diagram of the 65-bp indel variants. (**b**) Sequence peak diagram of the 65-bp indel variants.

**Figure 2 animals-10-00475-f002:**
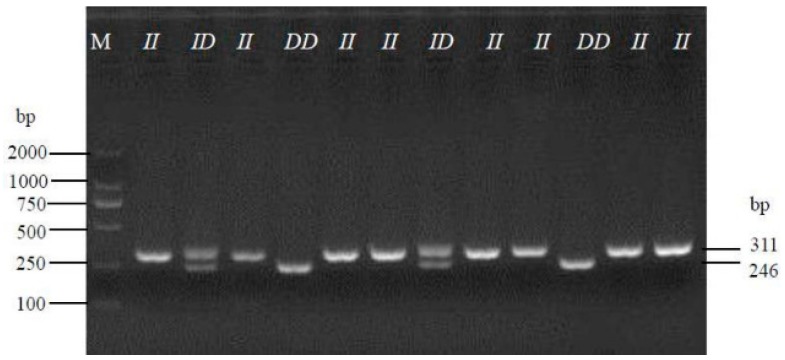
Electrophoresis pattern was performed to detect the 65-bp indel within the chicken GOLGB1 gene.

**Figure 3 animals-10-00475-f003:**
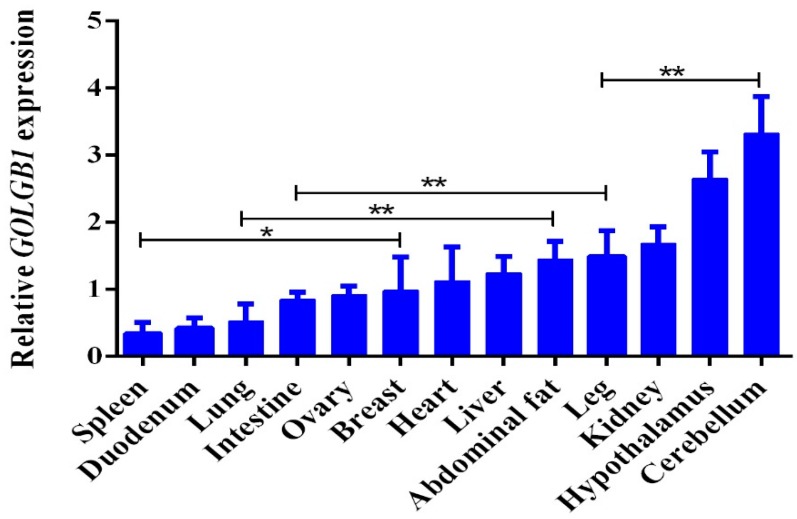
Relative mRNA expression levels of the *GOLGB1* gene in different tissues; the relative mRNA expression levels of *GOLGB1* were normalized to that of *β-actin*. *: represent a significant difference (*p* < 0.05); **: represent a very significant difference (*p* < 0.01).

**Figure 4 animals-10-00475-f004:**
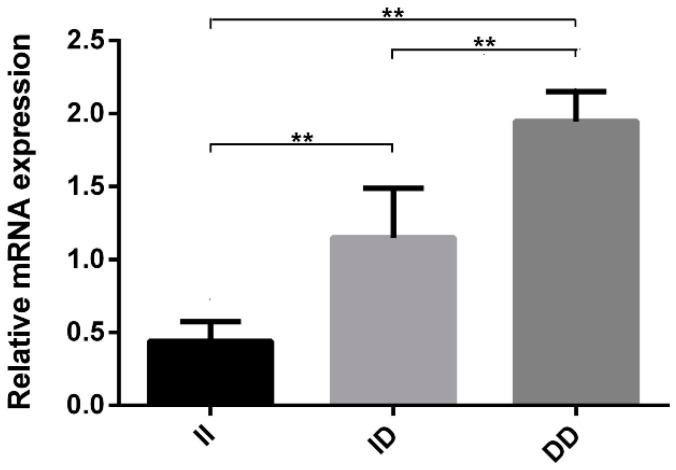
Expression levels of the *GOLGB1* gene in the livers of chickens with different genotypes. Data represent means ± SE; **: represent a very significant difference (*p* < 0.01).

**Table 1 animals-10-00475-t001:** Primers for amplifying the chicken *GOLGB1* gene.

Primer	Sequence (5′–3′)	Product Length (bp)
P1	F: TGTGGTAGCTCTCTCCTCCC	311
	R: AGGCTCTCCTGCTGACCATA	
P2	F: CACTGCGAACCCACGAGA	157
	R: CCCAAACCTGACAAACGGC	
*β-actin*	F: GACTGACCGCGTTACTCCCA	166
	R: CCAACCATCACACCCTGATGTC	

**Table 2 animals-10-00475-t002:** Genetic parameters of the *GOLGB1* 65-bp indel polymorphism in the eight chicken breeds.

Breeds	Genotypic Distribution				Allelic Frequencies		He	Ne	PIC	HWE (*p*-Value)
	*II*	*ID*	*DD*	*n*	*I*	*D*				
WC	49	30	9	88	0.73	0.27	0.397	1.658	0.318	0.124
SH	30	32	10	72	0.64	0.36	0.461	1.857	0.355	0.102
GS	37	27	5	74	0.68	0.32	0.392	1.646	0.315	0.125
XH	25	39	7	71	0.63	0.37	0.468	1.879	0.358	0.100
QY	26	27	7	60	0.66	0.34	0.450	1.818	0.349	0.998
LS	11	17	10	38	0.51	0.49	0.500	1.999	0.375	0.519
MH	335	211	32	578	0.76	0.24	0.363	1.569	0.297	0.871
N409	229	138	14	381	0.77	0.23	0.341	1.517	0.283	0.220

WC, Wenchang chickens; SH, Shanhuang chickens; GS, Gushi chicken; XH, Xinghua chickens; QY, Qingyuanma chickens; LS, Lushi chickens; MH, Mahuang chickens; N409, Tianlu yellow chickens. Ne, effective allele numbers; He, gene heterozygosity; PIC, polymorphic information content; *p*-value (HWE), *p*-value of Hardy-Weinberg equilibrium.

**Table 3 animals-10-00475-t003:** Association analysis of the *GOLGB1* 65-bp indel with N409 chicken growth traits.

Traits	(Mean ± SE)			*p*-Value
	*II*	*ID*	*DD*	
Body weight	1735.16 ± 12.21a	1705.38 ± 15.56a	1594.71 ± 49.077b	0.0121
Neck weight	117.17 ± 1.19a	111.80 ± 1.53b	104.86 ± 4.82b	0.0026
Abdominal fat weight	102.35 ± 1.91a	93.65 ± 2.46b	77.83 ± 7.74b	0.0006
Abdominal fat percentage	5.83 ± 0.089a	5.42 ± 0.11b	4.85 ± 0.36b	0.0015
Yellow index b of breast	3.78 ± 0.26a	3.71 ± 0.33a	7.36 ± 1.05b	0.0039
Eviscerated weight	1050.03 ± 7.96	1036.66 ± 10.22	998.77 ± 32.20	0.2188
Subcutaneous fat thickness	0.76 ± 0.11	0.76 ± 0.14	0.67 ± 0.44	0.1627
Chest width	7.09 ± 0.27	7.10 ± 0.35	7.12 ± 1.09	0.9565
Back width	8.62 ± 0.39	8.57 ± 0.50	8.40 ± 1.56	0.3255
Body length	43.07 ± 0.10	42.87 ± 0.13	42.71 ± 0.42	0.4130
Shank length	7.76 ± 0.23	7.80 ± 0.30	7.78 ± 0.94	0.6231
Cockscomb height	2.49 ± 0.35	2.38 ± 0.45	2.46 ± 1.41	0.1935
Yellow index L of abdominal fat	42.20 ± 0.30	43.03 ± 0.39	41.75 ± 1.23	0.2078
Yellow index L of breast	57.78 ± 0.36	56.92 ± 0.46	56.37 ± 1.46	0.2671
Yellow index a of breast	2.73 ± 0.08	2.60 ± 0.11	2.89 ± 0.34	0.5423
Yellow index L of leg	62.70 ± 0.24	62.53 ± 0.31	62.20 ± 0.98	0.4124
Yellow index a of leg	13.72 ± 0.19	13.64 ± 0.24	14.46 ± 0.76	0.5917
Yellow index b of leg	35.22 ± 0.31	34.84 ± 0.39	36.51 ± 1.23	0.3870

Different letters in the same row indicate a statistically significant difference (*p* < 0.05).

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
