# Peer review of "A Novel 65-bp Indel in the GOLGB1 Gene Is Associated with Chicken Growth and Carcass Traits"

_animals, 2020, doi:10.3390/ani10030475_

Round 1

Reviewer 1 Report

Authors did not incorporate all suggested modifications in the manuscript, there are still some modifications that need to be addressed.

· Lines 60-61: Please improve this sentence, this statement does not make sense. The identification of indels did not result in great progress, this is not the case, this is an exaggeration in terms of breeding programs. The indels did not “emerge”, they were identified by sequencing. Suggestion: The identification of thousands of indels in the last … years helped us to make progress in animal genetics and breeding.

· Lines 62-63: Please include few publications of identification of indels in the whole chicken genome and give some idea of the number of indels present in the chicken genome (or density), for example, you can cite the latest 2 publication about it. This will give some context for the reader. Of course, authors do not need to report all publication about that.

· Please provide the number of indels that were previously identified in this particular gene. You can find this information in public databases of genetic variants, such as https://ensembl.org/Gallus_gallus/Gene/Variation_Gene/Table?align=1760;db=core;g=ENSGALG00000041494;r=1:323123-357594.

· Lines 61-62 –Please cite the original scientific paper reporting the indel size, the paper cited is not a good example for that.

· Authors said that they submitted this indel in the EVA database, so please include this information in the manuscript, and also the provisional accession number provided by EVA team.

· Lines 105-106 – please provide name of the company – Illumina? Also, add more details about the sequencing such as the average of sequencing coverage for the 2 lines sequenced; sequencing library used, and length of the reads sequenced. Sequencing information should be placed after the DNA quality details. Also, which tool was used to identify the indels from the sequencing data? Please provide this information. Maybe authors can create a new subsection with sequencing details.

Author Response

Response to Reviewer 1

Thank you very much for your kind comments on our manuscript. Based on your comments, we have further revised and improved the manuscript.

Point 1: Lines 60-61: Please improve this sentence, this statement does not make sense. The identification of indels did not result in great progress, this is not the case, this is an exaggeration in terms of breeding programs. The indels did not “emerge”, they were identified by sequencing. Suggestion: The identification of thousands of indels in the last … years helped us to make progress in animal genetics and breeding.

Response: Thank you for your comments. Based on your suggestions, we have changed it to “The identification of thousands of indels in the last twenty years helped us to make progress in animal genetics and breeding”.

Point 2: Lines 62-63: Please include few publications of identification of indels in the whole chicken genome and give some idea of the number of indels present in the chicken genome (or density), for example, you can cite the latest 2 publication about it. This will give some context for the reader. Of course, authors do not need to report all publication about that.

Response: Thank you for your comments. We have provided relevant publications based on your suggestions. The details are as follows:

Yang S , Shi Z , Xiaoqian O U , et al. Whole-genome resequencing reveals genetic indels of feathered-leg traits in domestic chickens. Journal of Genetics, 2019, 98(2):1-8.

Clarissa B, Almas A. G, Hannah K. R., et al. Detection and characterization of small insertion and deletion genetic variants in modern layer chicken genomes. BMC Genomics, 2015, 16(1):562..

Point 3: Please provide the number of indels that were previously identified in this particular gene. You can find this information in public databases of genetic variants, such as https://ensembl.org/Gallus_gallus/Gene/Variation_Gene/Table?align=1760; db=core; g=ENSGALG00000041494; r=1:323123-357594.

Response: Thank you for your comments. In the Ensembl database, a total of 188 indels were identified in the GOLGB1 gene of chickens (https://ensembl.org/Gallus_gallus/Gene/Variation_Gene/Table?align=1760; db=core; g=ENSGALG00000041494; r=1:323123-357594).

Point 4: Lines 61-62 –Please cite the original scientific paper reporting the indel size, the paper cited is not a good example for that.

Response: Thank you for your comments. We have modified it in the manuscript.

Point 5: Authors said that they submitted this indel in the EVA database, so please include this information in the manuscript, and also the provisional accession number provided by EVA team.

Response: Thank you for your comments. We have submitted this Indel to the EVA database on February 18th and added this information to the manuscript, but the upload process generally takes about 1-2 months. We will add this serial number in the later revision.

Point 6: Lines 105-106 – please provide name of the company – Illumina? Also, add more details about the sequencing such as the average of sequencing coverage for the 2 lines sequenced; sequencing library used, and length of the reads sequenced. Sequencing information should be placed after the DNA quality details. Also, which tool was used to identify the indels from the sequencing data? Please provide this information. Maybe authors can create a new subsection with sequencing details.

Response: Thank you for your comments. The average sequencing coverage of the 2 lines was 10X. High-quality sequencing libraries were constructed by stringently following the standard protocol of IlluminaTruSeq™ DNA preparation kit (Illumina, CA, USA), then were sequenced on Hiseq2000 platform. Additional information is not yet available, as these data have not been published, and our article focuses on the association analysis of this gene.

Reviewer 2 Report

The reviewers have addressed all of the points raised in the review. Some minor writing issues but these are likely to be taken care of during the final editing stage.

Author Response

    Thank you very much for your kind comments on our manuscript. Based on your comments, we have further revised and improved the manuscript.

This manuscript is a resubmission of an earlier submission. The following is a list of the peer review reports and author responses from that submission.

Round 1

Reviewer 1 Report

The present study found a novel 65-bp indel within GOLGB1 located in a growth QTL on chromosome 1 related to growth traits in chickens. Genetic parameters of the 65-bp indel in multiple Chinese native chickens were calculated. Association analysis showed that the 65-bp indel in GOLGB1was associated with growth and carcass traits. Expression profile and expression pattern of GOLGB1 in different tissues were analyzed. This study provides important genetic data for Chinese native chickens. The authors support their finddings with a screen on allele frequenceies in different breeds. The sequence variants could be used to improve the production performance of chicken by marker assisted selection in the future. Therefore, I would like to recommend this manuscript to be acceptable for publication after revision. My detailed comments are as follows:

Major comments

The language of ABSTRACT, INTRODUCTION, and MATERIALS AND METHODS should be improved by a native English speaker, especially.

More information is needed on the local varieties collected. What was the age? What was the sex?

Minor corrections

L91-93 the “n” represents the sample size, so the “n” should be italic. Replace in other locations as well.

L115, L118, "(ABI, SGD)", “(Beijing, China)” Please be consistent with the manufacturer information provided (e.g., state/province and country only or city, state/province, and country) for specialized reagents. Replace in other locations as well.

L129 "Taq" should be italic.

Line 154, "TGCCCAGCAAAAGTGAAGAGCCTCACTGAGCTGCCCAGTACTCACTGCTGCTCATCCTGCTGGTG" should be changed "tgcccagcaa aagtgaagag cctcactgag ctgcccagta ctcactgctg ctcatcctgc tggtg"

L157, “in 8 different Chinese local chicken”, L170 “the eight Chinese local chicken”, 8 or eight? please keep the format consistent.

L170, “polymorphism information content (PIC)” , there is no need for abbreviations. Because, they have been defined in firt time. Replace in other locations as well.

L174 in table 2, change “Genotype distribution” to “Genotypic distribution”, change “total” to “n”, “P-value” should be italic. In L177 and L178. Replace in other locations as well.

standard deviation in L122 or standard error in L151? the data are presented as the mean ± SD or the mean ± SE?

In table 3, breed? Sample size?

Author Response

Thank you very much for your kind comments on our manuscript. Based on your comments, we have further revised and improved the manuscript.

Point 1: The language of ABSTRACT, INTRODUCTION, and MATERIALS AND METHODS should be improved by a native English speaker, especially.

Response: Thank you for your comment. We have modified and improved these sections based on your suggestions.

Point 2: More information is needed on the local varieties collected. What was the age? What was the sex?

Response: Thank you for your comment. More information on the local varieties has been added to the materials and methods.

Point 3: L91-93 the “n” represents the sample size, so the “n” should be italic. Replace in other locations as well.

Response: Thank you for your correction. The “n” in this study that represents the sample size has been changed italic.

Point 4: L115, L118, "(ABI, SGD)", “(Beijing, China)” Please be consistent with the manufacturer information provided (e.g., state/province and country only or city, state/province, and country) for specialized reagents. Replace in other locations as well.

Response: Thank you for your comment. We have changed the (ABI, SGD)", “(Beijing, China) in L115, L118 as required, and also modified elsewhere.

Point 5: L129 "Taq" should be italic.

Response: Thank you for your correction. The “Taq” in L129 has been changed italic.

Point 6: Line 154, "TGCCCAGCAAAAGTGAAGAGCCTCACTGAGC TGCCCAG TACT CACTGCTG CTCATCCTGCTGGTG" should be changed "tgcccagcaa aagtgaagag cctcactgag ctgcccagta ctcactgctg ctcatcctgc tggtg".

Response: Thank you for your correction. We have changed "TGCC CAGCAAAAGTGAAGAGCCTCACTGAGC TGCCCAG TACT CACTGCTG CTCATCCTGCTGGTG" in line 154 to the "tgcccagcaaaagtgaagag cctcactgagctg cccagtactcactgctgctcatcctgctggtg".

Point 7: L157, “in 8 different Chinese local chicken”, L170 “the eight Chinese local chicken”, 8 or eight? Please keep the format consistent.

Response: Thank you for your comment. We have changed all the eight in "the eight Chinese local chicken" in the original manuscript to 8 in the revised one.

Point 8: L170, “polymorphism information content (PIC)”, there is no need for abbreviations. Because, they have been defined in first time. Replace in other locations as well.

Response: Thank you for your comment. We have removed the abbreviations for PIC in L170, and have made changes elsewhere.

Point 9: L174 in table 2, change “Genotype distribution” to “Genotypic distribution”, change “total” to “n”, “P-value” should be italic In L177 and L178. Replace in other locations as well.

Response: Thank you for your comment. The “Genotype distribution” in table 2 on L174 has been modified to “Genotypic distribution”, and “total” has been modified to “n”, “P-value” in L177, L178 and elsewhere has been changed to italic.

Point 10: Standard deviation in L122 or standard error in L151? The data are presented as the mean ± SD or the mean ± SE?

Response: Thank you for your comment. The data are presented as the mean ± SE, Standard deviation in L122 has been changed.

Point 11: In table 3, breed? Sample size?

Response: Thank you for your correction. The breed is N409 chicken with 95 d of age, and the sample size is 382.

Reviewer 2 Report

The article has a correct sequence and development and affects very interesting aspects for the poultry meat production industry. 

Author Response

    Thank you very much for your kind comments on our manuscript. Based on your comments, we have further revised and improved the manuscript.

Reviewer 3 Report

This manuscript reports the identification of an indel associated with production traits of indigenous chicken breeds. The work is of interest, but diminished by lack of clarity of the experimental methods shown in the methods and results. I request that editors review this manuscript to improve the English language style and believe this will significantly clarify many of the specific points I raise below.

Specific points:

  1. Major (and should be addressed prior to publication):

Methods – 18 traits were measured but these were not specifically mentioned. (Additional traits that had no association should be mentioned – perhaps in a supplemental table – as these negative results are also valuable).

It is not specifically mentioned what the P2 primer set is used for – if P1 amplifies the 65bp indel of GOLGB1, what is the second set for? Please clarify these.

While the Methods section has many technical details it is hard for the reader to read and understand precisely what was done and why. The section on genetic variation and phenotyping comes after RNA expression and should be moved to follow genomic DNA extraction & analyses, so that the reader can follow the logical progression.

Why were eight different breeds used, and why were these breeds selected? (Do they represent a mix of fast and slow growing lines?) Why were tissues harvested from MH breed and not others, and why were those specific tissues selected?

“Liver tissue from 12 indigenous XH chickens was used to compare the relative mRNA expression levels of different genotypes in GOLGB1 gene.” – Why not use the XH that you already have expression values for?

Results –

3.1 Identification of a novel GOLGB1 65-bp indel polymorphism. From the introduction I got the impression that this was previous work; this needs to be clarified for the reader. The identification and sequencing of the indel is not described in the methods.

Table 3. Please clarify if this table is based upon the results of all 8 breeds or only N409 data. (Text is ambiguous.) If the latter, any association with the other breeds (or lack thereof) should also be reported.

Is “MH” breed the same as “MH7” breed reported in Table 2? Why was the MH breed chosen to examine expression profiles of GOLGB1?

  1. Optional (may improve impact/style):

The authors make the assertion that chicken chr 1 is associated with “a large number of QTLs” but chr 1 is also the largest chicken chromosome – is this assertion still true when size corrected?

The authors state that function of GOLGB1 in chicken is unclear. This appears to be a well studied gene in human and mouse with clear orthologs in birds, so it is reasonable to assume functional orthology in chicken. I recommend that the authors summarize presumed functions in chicken and focus more on the indel they identified, and its potential for chicken production.

The discussion again focused too much on re-stating what is known about human GOLGB1; much more interesting for the reader is how the variation in GOLGB1 is distributed across different breeds, how it is associated with production traits and how this information can be applied. Revising the discussion around these points would add to the overall interest of these research findings.

  1. Minor revisions (that the authors can correct without further review)

Reference 15 is incomplete.

Primer sequences used for b-actin are not reported.

Author Response

Thank you very much for your kind comments on our manuscript. Based on your comments, we have further revised and improved the manuscript.

Point 1: Methods – 18 traits were measured but these were not specifically mentioned. (Additional traits that had no association should be mentioned – perhaps in a supplemental table – as these negative results are also valuable).

Response: Thank you for your comments. We have also added the additional traits to Table 3.

Point 2: It is not specifically mentioned what the P2 primer set is used for – if P1 amplifies the 65bp indel of GOLGB1, what is the second set for? Please clarify these.

Response: Thank you for your comments. P2 primer is used to detect the mRNA expression level of GOLGB1 gene.

Point 3: While the Methods section has many technical details it is hard for the reader to read and understand precisely what was done and why. The section on genetic variation and phenotyping comes after RNA expression and should be moved to follow genomic DNA extraction & analyses, so that the reader can follow the logical progression.

Response: Thank you for your comments. We have modified it based on your suggestions.

Point 4: Why were eight different breeds used, and why were these breeds selected? (Do they represent a mix of fast and slow growing lines?) Why were tissues harvested from MH breed and not others, and why were those specific tissues selected?

Response: Thank you for your comments. The 8 different breeds were selected to test whether the indel mutation of this gene is universal in Chinese chickens. And they are all Chinese native chicken breeds which are slow-growing lines. GOLGB1 gene may be a gene related to chicken growth traits, and its mRNA expression profile in local chicken breeds should not differ much. N409 is also a hybrid line (slow-growth line) of Guangxi Sanhuang chicken, so we don't think that the choice of MH chicken or N409 will affect the mRNA expression of GOLGB1 gene. In addition, we made mRNA expression profiles for 13 tissues, not specifically selected tissues.

Point 5: “Liver tissue from 12 indigenous XH chickens was used to compare the relative mRNA expression levels of different genotypes in GOLGB1 gene.” – Why not use the XH that you already have expression values for?

Response: Thank you for your comments. We didn’t have previously the expression values for XH chickens with different genotypes.

Point 6: 3.1 Identification of a novel GOLGB1 65-bp indel polymorphism. From the introduction I got the impression that this was previous work; this needs to be clarified for the reader. The identification and sequencing of the indel is not described in the methods.

Response: Thank you for your comments. We have clarified this information in the introduction and supplemented it in the methods.

Point 7: Table 3. Please clarify if this table is based upon the results of all 8 breeds or only N409 data. (Text is ambiguous.) If the latter, any association with the other breeds (or lack thereof) should also be reported.

Response: Thank you for your comments. Table 3 showed the data from the association analysis of the GOLGB1 65-bp indel with N409 chicken growth traits. And the other breeds are used to detect the genotype and allele distribution of the 65-bp indel in different breeds of chickens.

Point 8: Is “MH” breed the same as “MH7” breed reported in Table 2? Why was the MH breed chosen to examine expression profiles of GOLGB1?

Response: Thank you for your comments. “MH” breed is the same as “MH7” breed reported in Table 2. GOLGB1 gene may be a gene related to chicken growth traits, and its mRNA expression profile in local chicken breeds should be similar, so we chose the MH breed to examine expression profiles of GOLGB1.

Point 9: The authors make the assertion that chicken chr 1 is associated with “a large number of QTLs” but chr 1 is also the largest chicken chromosome – is this assertion still true when size corrected?

Response: Thank you for your comments. A previous study reported that a large number of QTLs on chicken chromosome 1 are related to the important economic traits such as growth (Xie et al., 2012).

Point 10: The authors state that function of GOLGB1 in chicken is unclear. This appears to be a well studied gene in human and mouse with clear orthologs in birds, so it is reasonable to assume functional orthology in chicken. I recommend that the authors summarize presumed functions in chicken and focus more on the indel they identified, and its potential for chicken production.

Response: Thank you for your comments. In this study, the 65-bp indel of the GOLGB1 gene was associated with chicken body weight, neck weight, abdominal fat weight, abdominal fat percentage and the yellow index b of breast. Chickens with II genotype had significantly better body weight than those with ID and DD genotypes. Besides, we also found an interesting phenomenon that the yellow index b of breast is higher in DD genotype, which is opposite to the body weight and abdominal fat weight. The relationship between abdominal fat weight and skin color needs further study.

Point 11: The discussion again focused too much on re-stating what is known about human GOLGB1; much more interesting for the reader is how the variation in GOLGB1 is distributed across different breeds, how it is associated with production traits and how this information can be applied. Revising the discussion around these points would add to the overall interest of these research findings.

Response: Thank you for your comments. The GOLGB1 gene’s function has been reported more in humans and is described in the article. In other species, the function of GOLGB1 gene has been reported less. In zebrafish, it is found that the GOLGB1 gene is related to its ciliary function (Bergen et al., 2017).

Point 12: Reference 15 is incomplete.

Response: Thank you for your comments. We have completed it.

Point 13: Primer sequences used for b-actin are not reported.

Response: Thank you for your comments. We have added the sequence information of the β-actin primer to Table 1.

Reviewer 4 Report

Major comments

This study is interesting because there are few association studies using indels in chickens, however, there are import questions and doubts that authors should address. The methodology used is very simple, just one indel was selected in one particular gene for association analyses. Why did the authors select this particular gene? It is not clear. This gene is well studied in humans, but not in chickens. Nowadays with reducing sequencing costs, we easily sequence the whole genome. Also, this novel indel is present only in Chinese local breeds?  This is not clear. Did authors check in public SNP databases the existence of this particular deletion in other chicken populations? And, finally, if this deletion is novel, has been submitted to a public SNP database?

This indel is located in an intron, so we cannot expect that major effects in phenotypes will occur. Why did authors select this indel for the association study? Why not select an indel in an exonic/coding region?

Is there any QTL located in the GOLGB1 gene?

It is not clear how the authors identified this novel deletion. The whole genome was sequenced? Or only one gene? What sequencer was used? Please add this information in the material and methods. How many chickens were sequenced to identify this particular deletion?

Discussion is very poor.  For example, discuss more expression results and add references and results of previous work (Lines 232-235), for example there are information about this gene expression in chickens in a public expression atlas (https://www.ebi.ac.uk/gxa/home). Also, function of GOLGB1 was not studied in chickens (lines 215-216), but it was studied in humans, for example, authors can discuss more function of this gene in other species. In addition, authors did not discuss importance of the traits that they identified the significant associations with the novel indel, for example, importance of fat deposition or body weight. And why neck weight is important?

My major concern is that authors claim that there is no report of indel in this gene in chicken (lines 75-76). Where did the authors look? Only published papers? What about public databases of SNPs? Authors should check SNPs located in this gene in public genetic variant databases. The main database to look for SNPs was dbSNP from NCBI, but recently dbSNP database hosts only human SNPs. All SNPs from dbSNP from other species were moved to other databases, such as EVA (European Variation Archive). I checked this particular gene on Ensembl database, and I found 4,173 genetic variants in this gene (GOLGB1 - ENSGALG00000041494) including 188 Indels in chickens (https://ensembl.org/Gallus_gallus/Gene/Variation_Gene/Image?align=1760;db=core;g=ENSGALG00000041494;r=1:323123-357594). These variants at Ensembl were imported from dbSNP version 150 [remapped to GRCg6a]. Authors should check the number of genetic variants in this gene in chicken, and also check if there are indels in the same position as the one identified in this manuscript.

For example, in the study of Boschiero et al. 2015 (BMC Genomics. 2015; 16: 562), they identified 883 K high quality Indels from the analysis of several layer chicken lines from diverse breeds. They submitted those indels to public databases (https://www.ncbi.nlm.nih.gov/projects/SNP/snp_viewBatch.cgi?sbid=1062064 and https://www.ebi.ac.uk/ena/data/view/PRJEB9374). Authors can check if the 65 bp indel described here was identified before in this major study.

Minor comments

·         Figure 3 – please order all genes by their expression values. Do the different colors in this figure have a specific meaning? If not, please use just one color for all genes. Also, provide statistical significance in this table/analyses like in the Figure 4.

·         Please explain the acronyms Indels when you first use it. Insertions and deletions (Indels).

·         Lines 14- 15 – the emergence of indels or the emergence of sequencing approaches or bioinformatics analysis able to detect indels in the whole genomes?

·         Lines 20-21 – Is the “yellow chicken population” for egg or meat production? Also, please provide more details for all 8 chicken populations used (meat or egg production? Commercial or experimental populations?) (lines 89-95).

·         Why do references in the text start with reference 18, not 1 (line 45)?

·         Lines 60-61. This statement is out of context “Many Chinese local chickens show slow-growing and low-producing performance, which is not conductive to the development of the poultry industry.” All local breeds evaluated in this study have a low performance? It is no clear. Please explain it better.

·         Lines 66-67 – please provide the total number of Indels already identified in the chicken genome.

·         Please provide genome assembly version used (GRCg6a?) (lines 75 and 126). And also, provide chromosome coordinates of this particular novel indel identified.

·         Lines 95-96 – please improve writing (were used?). The total of 18 traits were evaluated? So authors need to provide all 18 traits evaluated, just few were mentioned. For example, what are all the fatness traits evaluated?

·         Lines 114, 118 – use qPCR instead of QPCR.

·         Line 127 – What did authors mean by “the special DNA fragment”? Please clarify. These primers have a good specificity? I checked one and it blasted against lots of regions in the chicken genome.

·         Lines 137-138 – please change to “genotype and allele frequencies”. Please remove the statement “based on genetic knowledge”. Suggestion: “The genotype and allele frequencies of eight breeds were calculated and the Hardy…”.

·         Please italicized “P-value” (Table 2, lines 177, 178), and gene name (line 180).

·         Correct typos on Figure 3 and 4 (Figture) (lines 192, 203).

·         Correct typo (line 234, GOLGB1gene).

·         Line 223, “broken genes” is not a good way to explain, please re-write.

·         Lines 228-229 “In addition, SNPs on introns often alter mRNA levels by affecting transcription, RNA elongation, splicing, or maturation [32].” What about indels? Do indels affect transcription?

·         Line 235 – What is the “positive effect”? An increase in the body weight? Because, for example, a lower content in fat deposition is better for poultry production. So positive and negative effects are relative concepts.

·         Lines 234-235 “Additionally, GOLGB1gene was highly expressed in brain tissues.” The authors did not discussed this sentence, this is only Results. Why this information is important? And how this can be related with the association results obtained?

Author Response

Thank you very much for your kind comments on our manuscript. Based on your comments, we have further revised and improved the manuscript.

Point 1: This study is interesting because there are few association studies using indels in chickens, however, there are import questions and doubts that authors should address. The methodology used is very simple, just one indel was selected in one particular gene for association analyses. Why did the authors select this particular gene? It is not clear. This gene is well studied in humans, but not in chickens. Nowadays with reducing sequencing costs, we easily sequence the whole genome. Also, this novel indel is present only in Chinese local breeds?  This is not clear. Did authors check in public SNP databases the existence of this particular deletion in other chicken populations? And, finally, if this deletion is novel, has been submitted to a public SNP database?

Response: Thank you for the suggestions. In an our previous study identifying candidate genes underlying chicken yellow skin with resequencing data from Genbank, golgin subfamily B member 1 (GOLGB1) gene is found to be located on chromosome 1 and might be associated with yellow skin phenotype potentially (data not published). As the candidate gene underlying yellow skin may also be associated with chicken growth, we screened the variations of the gene using the resequencing data from Chinese local chicken breeds and Recessive White Rock chicken with various growing rates. In this previous study, a 65-bp indel was identified and its association with chicken growth and carcass traits was analyzed further in this study. As this indel is novel, we submitted it to the European Variation Archive, but data reviewing will take some time.

Point 2: This indel is located in an intron, so we cannot expect that major effects in phenotypes will occur. Why did authors select this indel for the association study? Why not select an indel in an exonic/coding region?

Response: Thank you for your comment. Considering that an indel in an intron might affect gene chicken growth through regulating gene expression, we selected this indel as candidate marker for association study. There is no indel found in an exon/coding region in our variation identification study.

Point 3: Is there any QTL located in the GOLGB1 gene?

Response: Thank you for your comment. There is no QTL located in this gene, however, there are several reported QTLs in chicken chromosome 1 controlling chicken growth traits (Xie et al., 2012).

Point 4: It is not clear how the authors identified this novel deletion. The whole genome was sequenced? Or only one gene? What sequencer was used? Please add this information in the material and methods. How many chickens were sequenced to identify this particular deletion?

Response: Thank you for your comment. We discovered the gene by sequencing the whole genome of 10 Xinghua chickens and 10 Recessive White Rock chickens using Hiseq 2500. And we have added this information to the material and methods.

Point 5: Discussion is very poor.  For example, discuss more expression results and add references and results of previous work (Lines 232-235), for example there are information about this gene expression in chickens in a public expression atlas (https://www.ebi.ac.uk/gxa/home). Also, function of GOLGB1 was not studied in chickens (lines 215-216), but it was studied in humans, for example, authors can discuss more function of this gene in other species. In addition, authors did not discuss importance of the traits that they identified the significant associations with the novel indel, for example, importance of fat deposition or body weight. And why neck weight is important?

Response: Thank you for your comments. Based on your suggestions, we have modified the discussion section carefully. We have also added a discussion of the correlation between this novel indel and chicken growth traits and carcass traits. The GOLGB1 gene’s function has been reported more in humans and is described in the article. In other species, the function of GOLGB1 gene has been reported less. In zebrafish, it is found that the GOLGB1 gene is related to its ciliary function (Bergen et al., 2017).

Point 6: My major concern is that authors claim that there is no report of indel in this gene in chicken (lines 75-76). Where did the authors look? Only published papers? What about public databases of SNPs? Authors should check SNPs located in this gene in public genetic variant databases. The main database to look for SNPs was dbSNP from NCBI, but recently dbSNP database hosts only human SNPs. All SNPs from dbSNP from other species were moved to other databases, such as EVA (European Variation Archive). I checked this particular gene on Ensembl database, and I found 4,173 genetic variants in this gene (GOLGB1 - ENSGALG00000041494) including 188 Indels in chickens (https://ensembl.org/Gallus_gallus/Gene/Variation _Gene/Image? align=1760; db=core; g=ENSGALG00000041494;r=1:323123 -357594). These variants at Ensembl were imported from dbSNP version 150 [remapped to GRCg6a]. Authors should check the number of genetic variants in this gene in chicken, and also check if there are indels in the same position as the one identified in this manuscript.

Response: Thank you for your comments. We have modified the sentence to be is no report of indel function in this gene in chicken. In addition, the Indel we obtained through resequencing has been submitted to the EVA database and is under review. We found the Indel mutation at this locus in 10 Chinese local chickens and 10 Recessive White Rock chickens.

Point 7: Figure 3 – please order all genes by their expression values. Do the different colors in this figure have a specific meaning? If not, please use just one color for all genes. Also, provide statistical significance in this table/analyses like in the Figure 4.

Response: Thank you for your comments. We have order all genes in Figure 3 by their expression values and changed all genes to one color. We have detected the mRNA expression levels of 13 tissues in GOLGB1. When using the pairwise comparison method for statistical analysis of differences, it is found that there are significant differences between too many tissues, which are not easy to label. So we used the one-way analysis of variance method to mark the significance of breast muscle, leg muscle, and abdominal fat traits in the figure 3.

Point 8: Please explain the acronyms Indels when you first use it. Insertions and deletions (Indels).

Response: Thank you for your comments. We have explained it when using the acronyms indel for the first time.

Point 9: Lines 14- 15 – the emergence of indels or the emergence of sequencing approaches or bioinformatics analysis able to detect indels in the whole genomes?

Response: Thank you for your comments. Whole-genome resequencing is capable of detecting Indel of the entire genome.

Point 10: Lines 20-21 – Is the “yellow chicken population” for egg or meat production? Also, please provide more details for all 8 chicken populations used (meat or egg production? Commercial or experimental populations?) (lines 89-95).

Response: Thank you for your comments. The “yellow chicken population” on lines 20-21 is used for meat production, all 8 chicken populations is used for meat production (lines 89-95). And we have made changes in the manuscript.

Point 11: Why do references in the text start with reference 18, not 1 (line 45)?

Response: Thank you for your correction. We have modified it so that it starts at 1.

Point 12: Lines 60-61. This statement is out of context “Many Chinese local chickens show slow-growing and low-producing performance, which is not conductive to the development of the poultry industry.” All local breeds evaluated in this study have a low performance? It is no clear. Please explain it better.

Response: Thank you for your comments. The Xinghua chickens, Qingyuan Partridge chickens, Lushi chickens, Gushi chickens and Wenchang chickens in this study have a low performance.

Point 13: Lines 66-67 – please provide the total number of Indels already identified in the chicken genome.

Response: Thank you for your comments. There are about 15 chicken indel articles have been reported in the past 5 years (https://www.ncbi.nlm.nih.gov/pubmed). Many Indels in the chicken genome obtained by sequencing still have many unknown functions, and some Indels obtained by resequencing have been verified to have false positives through amplification and verification. So, we have no way to give the exact number of Indels.

Point 14: Please provide genome assembly version used (GRCg6a?) (lines 75 and 126). And also, provide chromosome coordinates of this particular novel indel identified.

Response: Thank you for your correction. We have provided the used genome assembly version (GRCg6a) (lines 75 and 126). And the chromosome coordinates of this particular novel indel identified is provided in the manuscript.

Point 15: Lines 95-96 – please improve writing (were used?). The total of 18 traits were evaluated? So authors need to provide all 18 traits evaluated, just few were mentioned. For example, what are all the fatness traits evaluated?

Response: Thank you for your comments. There are a total of 18 traits were recorded in the N409 population, including carcass traits, body size traits and fatness traits. We conducted correlation analysis between these 18 traits and 65-bp indel, and added them to Table 3.

Point 16: Lines 114, 118 – use qPCR instead of QPCR.

Response: Thank you for your correction. The QPCR has been changed to the qPCR in lines 114, 118.

Point 17: Line 127 – What did authors mean by “the special DNA fragment”? Please clarify. These primers have a good specificity? I checked one and it blasted against lots of regions in the chicken genome.

Response: Thank you for your comments. Line 127-The "special DNA fragment" refers to the DNA fragment containing the 65-bp indel. These primers have good specificity, and the desired product can be amplified well through experiments.

Point 18: Lines 137-138 – please change to “genotype and allele frequencies”. Please remove the statement “based on genetic knowledge”. Suggestion: “The genotype and allele frequencies of eight breeds were calculated and the Hardy…”.

Response: Thank you for your comments. According to your suggesting, we have deleted the “based on genetic knowledge”. And this sentence has been changed to “The Genotypic and allele frequencies of the 8 breeds were calculated and the Hardy-Weinberg equilibrium (HWE) was calculated by using the SHEsis program”.

Point 19: Please italicized “P-value” (Table 2, lines 177, 178), and gene name (line 180).

Response: Thank you for your correction. We have changed the “P-value” (Table 2, lines 177, 178) and gene name (line 180) to italics.

Point 20: Correct typos on Figure 3 and 4 (Figture) (lines 192, 203).

Response: Thank you for your correction. The typos in Figures 3 and 4 (Figture) have been corrected (lines 192, 203).

Point 21: Correct typo (line 234, GOLGB1gene).

Response: Thank you for your correction. The typo (line 234, GOLGB1gene) has been corrected.

Point 22: Line 223, “broken genes” is not a good way to explain, please re-write.

Response: Thank you for your comments. According to your suggesting, we have changed the “broken genes” to the “non-coding spacer sequences that interrupt the linear expression of genes”.

Point 23: Lines 228-229 “In addition, SNPs on introns often alter mRNA levels by affecting transcription, RNA elongation, splicing, or maturation [32].” What about indels? Do indels affect transcription?

Response: Thank you for your comments. Indel can affect transcription. The following is the corresponding literature,

Xu, Y., Shi, T., Zhou, Y., Liu, M., Klaus, S., Lan, X., Chen, H. A novel PAX7 10-bp indel variant modulates promoter activity, gene expression and contributes to different phenotypes of Chinese cattle. Scientific reports, 2018.8(1), 1-10.

Cui, Y., Yan, H., Wang, K., Xu, H., Zhang, X., Zhu, H., Liu, J., Qu, L., Lan, X., and Pan, C. Insertion/Deletion Within the KDM6A Gene Is Significantly Associated With Litter Size in Goat. Frontiers in Genetics. 2018. 9- 91.

Point 24: Line 235 – What is the “positive effect”? An increase in the body weight? Because, for example, a lower content in fat deposition is better for poultry production. So positive and negative effects are relative concepts.

Response: Thank you for your comments. In this study, the 65-bp indel was significantly associated with body weight, abdominal fat weight and abdominal fat percentage, chickens with II genotype had significantly better body weight than those with ID and DD genotypes, and analysis of genetic parameters indicated that “I” was the predominant allele. So we argued that the 65-bp indel had a positive effect on chicken’s body weight, neck weight, abdominal fat weight and abdominal fat percentage.

Point 25: Lines 234-235 “Additionally, GOLGB1 gene was highly expressed in brain tissues.” The authors did not discuss this sentence, this is only Results. Why this information is important? And how this can be related with the association results obtained?

Response: Thank you for your comments. Based on your suggestion, we modified and discussed this result. GOLGB1 gene was highly expressed in cerebellum, hypothalamus, kidney, liver and abdominal fat, suggesting that it might be related to growth and fat deposition. Notably, the GOLGB1 gene with the DD genotype showed higher expression than the II and ID genotypes in the liver tissue of chickens. The body weight, neck weight, abdominal fat weight and abdominal fat percentage of individuals with DD genotype are lower than those with the genotype II. Therefore, we argued that the 65-bp indel had a positive effect on chicken’s body weight, neck weight, abdominal fat weight and abdominal fat percentage.